Benchmark datasets for SARS-CoV-2 surveillance bioinformatics

Xiaoli Lingzi 1
Hagey Jill V. 1
Park Daniel J. 2
Gulvik Christopher A. 1
Young Erin L. 3
Alikhan Nabil-Fareed 4
Lawsin Adrian 1
Hassell Norman 1
Knipe Kristen 1
Oakeson Kelly F. 3
Retchless Adam C. 1
Shakya Migun 5
Lo Chien-Chi 5
Chain Patrick 5
Page Andrew J. 4
Metcalf Benjamin J. 1
Su Michelle 1
Rowell Jessica 6
Vidyaprakash Eshaw 6
Paden Clinton R. 1
Huang Andrew D. 6
Roellig Dawn 1
Patel Ketan 1
Winglee Kathryn 1
Weigand Michael R. yrh8@cdc.gov 1
Katz Lee S. gzu2@cdc.gov 1
1 Strain Surveillance and Emerging Variant Team, Centers for Disease Control and Prevention , Atlanta , GA , United States of America
2 Broad Institute of MIT and Harvard , Cambridge , MA , United States of America
3 Utah Public Health Laboratory , Salt Lake City , UT , United States of America
4 Quadram Institute Bioscience , Norwich Research Park , Norwich , United Kingdom
5 Bioscience Division, Los Alamos National Laboratory , Los Alamos , NM , United States of America
6 SARS-CoV-2 Emerging Variant Sequencing Project Dry Lab Group Laboratory and Testing Task Force COVID-19 Emergency Response, Centers for Disease Control and Prevention , Atlanta , GA , United States of America
Kant Ravi
Electronic publication date: 2022 Sep 5
Publication date: 2022
Volume: 10
Electronic Location ID: e13821
Received 2022 Mar 31; Accepted 2022 Jul 8
Copyright: ©2022 Xiaoli et al.
Copyright year: 2022
Copyright holder: Xiaoli et al.
License: This is an open access article distributed under the terms of the Creative Commons Attribution License, which permits unrestricted use, distribution, reproduction and adaptation in any medium and for any purpose provided that it is properly attributed. For attribution, the original author(s), title, publication source (PeerJ) and either DOI or URL of the article must be cited.
License URL: https://creativecommons.org/licenses/by/4.0/

Keywords: Standardization, sha256, Benchmarking, WGS, COVID-19

Funding: Biotechnology and Biological Sciences Research Council (BBSRC) BBSRC Institute Strategic Programme Microbes in the Food Chain Quadram Institute Bioscience BBSRC BB/CCG1860/1 National Institute of Allergy and Infectious Diseases U19AI110818 Bill and Melinda Gates Foundation INV-002717 Federal Appropriations to the Centers for Disease Control and Prevention Andrew J. Page and Nabil-Fareed Alikhan were supported by the Biotechnology and Biological Sciences Research Council (BBSRC); their research was funded by the BBSRC Institute Strategic Programme Microbes in the Food Chain BB/R012504/1 and its constituent project BBS/E/F/000PR10352, also Quadram Institute Bioscience BBSRC funded Core Capability Grant (project number BB/CCG1860/1). Daniel J. Park was supported by the National Institute of Allergy and Infectious Diseases (U19AI110818) and the Bill and Melinda Gates Foundation (INV-002717). Lingzi Xiaoli, Jill V. Hagey, Chris Gulvik, Adrian Lawsin, Norman Hassell, Kristen Knipe, Adam C. Retchless, Benjamin J. Metcalf, Michelle Su, Clinton R. Paden, Andrew D. Huang, Dawn Roeillig, Ketan Patel, Kathryn Winglee, Michael R. Weigand, and Lee S. Katz were funded by Federal Appropriations to the Centers for Disease Control and Prevention. The funders had no role in study design, data collection and analysis, decision to publish, or preparation of the manuscript.

==============================
Background

Severe acute respiratory syndrome coronavirus 2 (SARS-CoV-2), the cause of coronavirus disease 2019 (COVID-19), has spread globally and is being surveilled with an international genome sequencing effort. Surveillance consists of sample acquisition, library preparation, and whole genome sequencing. This has necessitated a classification scheme detailing Variants of Concern (VOC) and Variants of Interest (VOI), and the rapid expansion of bioinformatics tools for sequence analysis. These bioinformatic tools are means for major actionable results: maintaining quality assurance and checks, defining population structure, performing genomic epidemiology, and inferring lineage to allow reliable and actionable identification and classification. Additionally, the pandemic has required public health laboratories to reach high throughput proficiency in sequencing library preparation and downstream data analysis rapidly. However, both processes can be limited by a lack of a standardized sequence dataset.

Methods

We identified six SARS-CoV-2 sequence datasets from recent publications, public databases and internal resources. In addition, we created a method to mine public databases to identify representative genomes for these datasets. Using this novel method, we identified several genomes as either VOI/VOC representatives or non-VOI/VOC representatives. To describe each dataset, we utilized a previously published datasets format, which describes accession information and whole dataset information. Additionally, a script from the same publication has been enhanced to download and verify all data from this study.

Results

The benchmark datasets focus on the two most widely used sequencing platforms: long read sequencing data from the Oxford Nanopore Technologies platform and short read sequencing data from the Illumina platform. There are six datasets: three were derived from recent publications; two were derived from data mining public databases to answer common questions not covered by published datasets; one unique dataset representing common sequence failures was obtained by rigorously scrutinizing data that did not pass quality checks. The dataset summary table, data mining script and quality control (QC) values for all sequence data are publicly available on GitHub: https://github.com/CDCgov/datasets-sars-cov-2.

Discussion

The datasets presented here were generated to help public health laboratories build sequencing and bioinformatics capacity, benchmark different workflows and pipelines, and calibrate QC thresholds to ensure sequencing quality. Together, improvements in these areas support accurate and timely outbreak investigation and surveillance, providing actionable data for pandemic management. Furthermore, these publicly available and standardized benchmark data will facilitate the development and adjudication of new pipelines.

Introduction

The severe acute respiratory syndrome coronavirus 2 (SARS-CoV-2), which caused the coronavirus disease 2019 (COVID-19) pandemic, has infected more than 456 million people globally resulting in over six million deaths as of March 14th, 2022 (https://covid19.who.int/). Since the first genomic sequence of SARS-CoV-2 was made publicly available on January 10th, 2020, whole genome sequencing (WGS) and bioinformatics analyses have been performed extensively to characterize and surveil the virus’s evolution. The SARS-CoV-2 virus has undergone rapid evolutionary expansion, leading to the emergence of discrete variants, some of which exhibit altered infectivity, clinical severity, or decreased susceptibility to medical treatments (Otto et al., 2021; Pascall et al., 2021; Abdool Karim & De Oliveira, 2021). The observed waves of novel variants of SARS-CoV-2, with greater transmissibility than the original strain, including B.1.1.7 (Alpha) and B.1.617.2 (Delta), B.1.1.529 (Omicron) has emphasized the need for real-time sequence-based virus surveillance (Davies et al., 2021; Elliott et al., 2021). Such surveillance has its roots in genomic epidemiology, which had already shown utility and impact for surveillance of other infectious disease agents including but not limited to bacterial foodborne organisms (PulseNet, 2016), influenza (Shu & McCauley, 2017), ebola (Quick et al., 2016), and norovirus (Vega et al., 2011).

In 2020, many national viral genomics consortia were established to coordinate SARS-CoV-2 genome sequencing support of public health response efforts including: the Coronavirus Disease 2019 (COVID-19) Genomics UK Consortium (COGUK, 2020), SARS-CoV-2 Sequencing for Public Health Emergency Response, Epidemiology and Surveillance (SPHERES) in the USA (CDC, 2020a; CDC, 2020b; CDC, 2020c), the Canadian COVID Genomics Network (CanCOGeN, 2020), and the Indian SARS-CoV-2 Genomics Consortium (INSACOG, 2020). Such consortia require substantial economic, trained personnel, equipment, and technical resources, which present considerable barriers to many countries, but are necessary for global control of SARS-CoV-2 outbreaks (Helmy, Awad & Mosa, 2016; Brito et al., 2021; Chen et al., 2022). Leadership from these consortia in their respective countries led many public health laboratories to start building sequencing and bioinformatics capacity for real-time genomic surveillance of SARS-CoV-2. One central challenge to coordinating these efforts, which is not unique to SARS-CoV-2 surveillance, has been the diversity of high-throughput sequencing platforms employed by various laboratories, varying sample preparation methods, and different amplicon strategies. The sequencing platforms include Oxford Nanopore, Illumina, Pacific Biosciences, Ion Torrent; sample preparation methods include amplicon-based, shot-gun, and metagenomics; amplicon strategies include ARTIC V3, ARTIC V4, and SWIFT. Therefore, it is necessary to evaluate whether the sequencing platform that generates data influences downstream bioinformatic processing of consensus sequences in any use-cases, e.g., genomic surveillance or epidemiology. Additionally, the sample preparation methods differ widely, including metagenomic, amplicon and hybrid capture, each of which may impact the creation of assembled sequences or consensus sequences.

Standardization is further complicated by the large number of bioinformatics applications that were either expanded or developed from scratch to rapidly meet the needs of the COVID-19 pandemic. For genomic epidemiology, these platforms include Augur, Auspice, and UShER (Turakhia et al., 2021; Hadfield et al., 2018). For lineage detection, these platforms include Pangolin and Nextclade (O’Toole et al., 2021; Hadfield et al., 2018). For QC, many pipelines use different, modular combinations of open-source software including FastQC, NCBI human scrubber, Kraken, Trimmomatic, BBDuk, SeqyClean, SAMtools, and Viral Annotation DefineR (VADR), and Artic field bioinformatics (Schäffer et al., 2020; Li et al., 2009; Zhbannikov et al., 2017; BBMap, 2021; Bolger, Lohse & Usadel, 2014; Wood & Salzberg, 2014; Katz et al., 2021; Andrews, 2010; ARTIC, 2020). A more comprehensive review of the variety of software used was recently published by Hu et al. (2021). There are also many instances of commercial software and/or closed source packages that were rapidly developed, including Illumina’s DRAGEN, Clear Dx™ WGS SARS-CoV-2 Bioinformatics Pipeline (BIP), EPISEQ SARS-COV-2 (bioMérieux, Marcy-l’Étoile, France), and CLC Genomics Workbench (QIAGEN, Hilden, Germany).

Critically, a common set of sequence data has not been staged for all available bioinformatics applications to compare their performance. Furthermore, laboratories lack standardized data with which to test their competencies in these analyses, or even train personnel, which is a common requirement for compliance with quality management systems.

To address this gap, we propose specific SARS-CoV-2 sequencing benchmark datasets to aid laboratories in building bioinformatics infrastructure, validating cross-platform sequence analyses, evaluating bioinformatics pipelines, and verifying QC procedures. These datasets may also serve as a training or competency resource for new laboratory staff to understand features of sequence data that either pass or fail common QC metrics.

Materials & Methods

Datasets

Each dataset listed in Table 1 was designed to address a specific need, following the datasets format described in Timme et al. (2017). Briefly, the tab-separated file format organizes information into two sections: the header and the data. The header contains metadata describing the whole dataset including a unique name, the source of data, and the intended use of the dataset. Directly following the header, the data section includes sample-specific information such as sample name, NCBI accessions, cluster information, and hashsum values of each sequencing read file to be downloaded. This second section may also contain additional, optional columns (such as GISAID accession, lineage, or amplicon strategy) relevant to individual datasets. Essentially, all FASTQ files for sequences in the datasets are stored on NCBI SRA while a simple spreadsheet with accessions is stored in our software repository (Cock et al., 2010). A summary of the entire workflow used to identify and validate sequences to be included in the datasets is provided in Fig. 1.

Download script

The script to download each dataset is called GenFSGopher.pl, as described in Timme et al. (2017). GenFSGopher.pl reads a spreadsheet of the user’s choosing and downloads the associated data from the specified NCBI accession. Spreadsheets for each dataset are available in the datasets folder of the repository. Each file downloaded by the script is checked against its unique identifier of hashsum, ensuring that the data downloaded on one computer is exactly the same as another computer.

QC metrics and thresholds

Here, we defined specific metrics for both the raw reads and resulting consensus assembly that each sequence must pass for inclusion to our benchmark datasets. There are differences in QC metric thresholds set by pipelines and between the major public repositories of SARS-CoV-2 which cannot be easily accounted for. We summarized the most common metric cutoffs used for our purposes in Table 2.

Table 1 The summary description for six datasets. Each dataset is numbered, named, and given a description. The intended use is also listed.

Dataset	Name	Description	Intended use	Reference	
1	Boston outbreak	A cohort of 63 samples from a real outbreak with three introductions, metagenomic approach	To understand the features of virus transmission during a real outbreak setting	Lemieux et al. (2021)	
2	CoronaHiT rapid	A cohort of 39 samples prepared by 18 h wet-lab protocol and sequenced by two platforms (Illumina vs MinION), amplicon-based approach	To verify that a bioinformatics pipeline finds virtually no differences between sequences from the same genome run on different platforms.	Baker et al. (2021)	
3	CoronaHiT routine	A cohort of 69 samples prepared by 30 h wet-lab protocol and sequenced by two platforms (Illumina vs MinION), amplicon-based approach	To verify that a bioinformatics pipeline finds virtually no differences between sequences from the same genome run on different platforms.	Baker et al. (2021)	
4	VOI/VOC lineages	A cohort of 16 samples from 11 representative CDC defined VOI/VOCa lineages as of 05/30/2021, amplicon-based approach	To benchmark lineage-calling bioinformatics software, especially for VOI/VOCs.	This study	
5	Non-VOI/VOC lineages	A cohort of 39 samples from representative non VOI/VOCa lineages, amplicon-based approach	To benchmark lineage-calling bioinformatics software, nonspecific to VOI/VOCs.	This study	
6	Failed QC	A cohort of 24 samples failed basic QC metrics, covering 8 possible failure scenarios, amplicon-based approach	To serve as controls to test bioinformatics QC cutoffs.	This study	
Notes.

a VOI, variant of interest; VOC, variant of concern

Figure 1 Automated workflow for identifying representative sequences for datasets.

Sequences go through several quality checks before being considered as part of a benchmark dataset. These checks include lineage agreement with Pangolin, a minimum Phred score, a minimum depth of coverage, a check with the software TheiaCov, a check of the amplicon strategy, a minimization of the count of SNPs in regards to a reference genome, and a check against the spike region’s mutations. Asterisks denote steps taken with in-house python scripts.

Sequence quality evaluation

We evaluated the quality of the sequences for the six datasets through three steps. In the first step, we used FastQC to evaluate the basic read quality of the FASTQ files (Fig. 1, Table 2: 1–3). In the second step, we used the genome assembly of Wuhan-1 (accession number: NC_045512.2) as the reference and evaluated the sequencing depth per nucleotide across the total length of the Wuhan reference (Fig. 1, Table 2: 4–8). Metrics 4–6 in Table 2 indicated the average depth per nucleotide, the variation of depth as well as the degree of depth variation. Metrics 7–8 in Table 2 were used to estimate the number of ambiguous nucleotides that would be observed in the downstream consensus assembly. In the third step, we tested all six datasets using the TheiaCoV (formerly ‘Titan’) workflow (v1.4.4) (Libuit et al., 2022) with the default UShER (v0.3.0) as the inference engine (Turakhia et al., 2021). TheiaCov is included in our existing bioinformatic infrastructure, is already being used by some public health laboratories, reports our determined metrics of interest, and summarizes additional QC metrics (Table 2: 9–19) to indicate four aspects: (a) input data size after trimming and human read removal, (b) conflicting read taxonomy, i.e., contamination, (c) amino acid changes especially spike protein mutations, (d) lineage or clade information (Fig. 1). Similar to most SARS-CoV-2 workflows, Pangolin is incorporated into TheiaCov.

Table 2 QC metrics.

QC metrics are shown with their thresholds, which bioinformatics tool we used, and the QC category.

No.	QC Metrics	Cutoff	Tool (version)	Category	
1	total reads	NC	FastQC	Step 1: Fastq quality check	
2	read length	NC	FastQC	Step 1: Fastq quality check	
3	average phred score	>25	FastQC	Step 1: Fastq quality check	
4	mean depth per nucleotide (MDN)	>10	Samtools	Step 2: Depth check	
5	standard deviation for MDN	NC	Samtools	Step 2: Depth check	
6	coefficient of variation for MDN	NC	Samtools	Step 2: Depth check	
7	number of nucleotides with depth <10 (for Illumina)	<3000	Samtools	Step 2: Depth check	
8	number of nucleotides with depth <20(for nanopore)	<3000	Samtools	Step 2: Depth check	
9	number of paired-end reads	NC	Titan 1.4.4	Step 3: Bioinformatics workflow check	
10	assembly total length	>29400	Titan 1.4.4	Step 3: Bioinformatics workflow check	
11	ambiguous Ns	<10%	Titan 1.4.4	Step 3: Bioinformatics workflow check	
12	assembly mean coverage	>25	Titan 1.4.4	Step 3: Bioinformatics workflow check	
13	% mapped to the Wuhan reference	>65%	Titan 1.4.4	Step 3: Bioinformatics workflow check	
14	VADR alert number	<=1	Titan 1.4.4	Step 3: Bioinformatics workflow check	
15	nextclade_aa_dels	NC	Titan 1.4.4	Step 3: Bioinformatics workflow check	
16	nextclade_aa_subs	NC	Titan 1.4.4	Step 3: Bioinformatics workflow check	
17	nextclade_version	NC	Titan 1.4.4	Step 3: Bioinformatics workflow check	
18	pango_lineage	NC	Titan 1.4.4	Step 3: Bioinformatics workflow check	
19	pangolin_version	NC	Titan 1.4.4	Step 3: Bioinformatics workflow check	
Notes.

NC not a criterion

These values are reported but not used as criteria for passing or failing a sample.

Dataset 1—an outbreak

This dataset describes an outbreak resulting from three independent introductions of SARS-CoV-2 in a large metropolitan city (Lemieux et al., 2021). The intended use of these two datasets is to evaluate methods for phylogenetic reconstruction, as the resulting phylogenetic tree should accurately delineate three clusters and an outgroup. The expected tree, placement of each sequence and the outgroup are labeled in the “tree” row of the dataset table. It is important to note this study was conducted early in the COVID-19 pandemic, prior to the widespread adoption of amplicon-based sequencing (ARTIC, Swift, etc.), and therefore utilized a shotgun metagenomic approach.

Datasets 2 and 3—multiple platforms

We selected these data from Baker et al. (2021), which describes a novel library preparation method for SARS-CoV-2 for sequencing called Coronavirus High Throughput (CoronaHiT). CoronaHiT can provide flexible throughput using either Illumina or Nanopore technology, which allows sequencing up to 96 samples on Nanopore or 2880 samples on Illumina in a single experiment and generating more even coverage between multiplexed samples. Following the CoronaHiT library preparation, the genomes were sequenced in parallel with both the Illumina and Oxford Nanopore Technologies (ONT) platform. The authors also supply genome sequence data from the ONT platform using the standard library preparation method, known as “LoCost” that can sequence 11-95 samples with one negative control using the Native Barcoding Expansion 96 kit (Quick, 2020). Therefore, each genome in datasets 2 and 3 has been sequenced by three approaches (Illumina CoronaHiT, Nanopore CoronaHiT and Nanopore LoCost). Baker et al. (2021) further described separate “rapid” and “routine” methods for CoronaHiT, which are also reflected in datasets 2 and 3, respectively. The difference between “rapid” and “routine” methods is the run time when using MinION. The “rapid” version ran for 18 hrs, while “routine” version ran for 30 hrs. The intended use of these two datasets is to verify the consistency of bioinformatics applications for generating consensus sequences from input data produced with various library strategies or sequencing technology.

Dataset 4—lineages

This dataset contains one genome per focal lineage of SARS-CoV-2, named according to the PANGO nomenclature. Important lineages in this paper are defined as a CDC-specified variant of concern (VOC) or variant of interest (VOI) lineage as of June 15th, 2021 (“SARS-CoV−2 Variant Classifications and Definitions”, 2021). At that time, the list included 10 PANGO lineages: six VOCs (B.1.1.7, B.1.351, B.1.427, B.1.429, B.1.617.2, P.1) and four VOIs (B.1.525, B.1.526, P.2, B.1.617.1). Using CDC internally curated consensus sequences (Table S1) for each lineage as the reference, we developed an automatic workflow to select representative Illumina paired-end reads generated with ARTIC V3 primers from NCBI SRA that satisfied our QC metric cutoffs. See the automated data mining workflow section below for more details. Additionally, we verified that a corresponding consensus sequence record was also present in GISAID EpiCov. This dataset is intended for benchmarking PANGO lineage-assignment pipelines, particularly for those classified as VOI/VOC lineages.

Dataset 5—more lineages

This dataset contains a complementary selection of 39 non-VOI/non-VOC lineages, chosen from a collection of CDC internally curated consensus sequences. In addition to raw sequencing reads, accessions to the consensus genome sequences available in both GISAID and NCBI Reference Sequence Database (RefSeq) for each sample in this dataset are provided. The intended use for this dataset is to benchmark lineage-assignment bioinformatics pipelines, nonspecific to VOI/VOCs. Lineages were assigned to all sequences using the Pangolin v3.1.3 classification software (O’Toole et al., 2021). All sequences were aligned to a SARS-CoV-2 reference genome (NCBI accession number: MT019531) using the SSW library, an extension of Farrar’s Striped Smith-Waterman algorithm (Zhao et al., 2013). Representative sequences for each lineage were obtained to avoid artifacts introduced from the aligned and classified sequences by taking the earliest sample from the most abundant genome alignment profile (sorted by genome length) per lineage.

Dataset 6—QC failures

Sequence data that failed at least one QC metric described below were manually selected for this dataset from a large collection of sequence runs performed by the CDC and Utah Public Health Laboratory for national surveillance. These QC failures include low Phred score, low coverage breadth, low mean coverage depth, human sequence contamination, long stretches of ambiguous nucleotides in the consensus (>100 Ns in a row), and amplicon dropout. All human contamination was anonymized by replacing patient read data with the corresponding sequence from a published reference. Briefly, reads containing human sequences were first separated using NCBI’s Human Read Removal Tool (Katz et al., 2021) and mapped to the T2T reference genome (Nurk et al., 2022) using bowtie2 (Langmead & Salzberg, 2012). The sequences of mapped read pairs were then replaced with aligned reference sequence and the anonymized reads subsequently recombined with the remaining non-human reads. In this way, Personal Identifiable Information (PII) has been removed from the dataset while retaining observed frequency and location of human reads. The intended use of this dataset is to help calibrate bioinformatics pipelines against common quality control thresholds for analyses or submission to public data repositories.

Automated data mining workflow

To find genomes that met eligibility criteria for the VOC/VOI dataset, a custom workflow which included in-house python scripts was developed, noted with an asterisk in Fig. 1. To begin, 383,000 consensus genomes from the GISAID database assigned to each designated VOC/VOI were downloaded on April 23rd, 2021. Initially, Pangolin (v2.4.2) was run to confirm the lineage was unchanged from the time of submission using PangoLEARN (container dated 2021-05-19).

For each VOC/VOI lineage, a multi-FASTA file was created by filtering the original file containing 383,000 sequences using seqkit with the subcommand grep (v1.0) (Shen et al., 2016). Next, Snippy (v4.3.8) was run on the multi-FASTA file against the internal CDC reference sequence (Table S1). These data mining bioinformatics workflows were incorporated on our GitHub repository. For each VOI/VOC lineage, the workflow output four metrics: number of single nucleotide polymorphisms (SNPs) and LowCov for each individual sequence, a SNP range and LowCov range for the multi-FASTA (Seemann, 2019). LowCov indicated the number of low coverage sites with a depth cutoff of 10. We selected the 5 samples for each VOI/VOC lineage whose GISAID consensuses have the fewest number of SNPs and LowCov from the corresponding reference. We linked the selected GISAID consensus to the NCBI Sequence Read Archive (SRA) accession. Next, the raw reads from each sample’s SRA accession were run through TheiaCov v1.4.4, using the Terra.bio interface, to ensure that they met our minimum QC thresholds (Table S2). As part of TheiaCov’s workflow, Pangolin (v3.1.3; container dated 2021-06-15) was rerun on samples with UShER (v0.3.1) being used as the inference engine for lineage calls. This version of Pangolin notably utilizes Scorpio (v0.3.1) and Constellations (v0.0.5). If the read sequences failed any of our QC thresholds, we removed them from consideration until we found the most representative genome for each VOI/VOC lineage. In some cases, we had to analyze all publicly available sequences for a specific lineage to meet our goal.

We also confirmed that the selected SARS-CoV-2 sequences were paired-end, sequenced by the Illumina platform, and used ARTIC V3 primers for amplification through querying their SRA run accessions in the NCBI SRA page. Instead of manually checking this information, we generated another automatic process using the browser automation package Selenium (v3.141.0) for Python3 to harvest the pertinent information (Muthukadan, 2018). This script, named NCBI_Scraping.py, accounted for differences in the location and description of the construction protocol (i.e., Artic protocol V3, ARTIC V3 PCR-tiling of viral cDNA, ARTIC v3 amplicons etc.). From this process a CSV file containing a list of SRR accessions that met our criteria was generated. The methods for our entire automated data mining workflow are encapsulated in scripts that are available in the project repository including a readme.md file describing how to use them.

Results and Discussion

Datasets

We provide the community with six benchmark datasets of SARS-CoV-2 genomic sequencing data that can be used for a variety of applications (Table 1). These curated datasets consist of a defined outbreak (dataset 1), different sequencing approaches and platforms (dataset 2 and 3), different lineages (datasets 4 and 5), and commonly encountered sequencing failures (dataset 6). We also provide a script that downloads the data behind these datasets. Our benchmark datasets can be applied to many different use-cases, such as bioinformatics pipeline evaluation, QC verification, cross sequencing platform validation, personnel training, or competency testing. We hope this effort will facilitate public health laboratories at different development stages to build robust sequencing and bioinformatics infrastructure for accurate real-time sequence-based virus outbreak investigation and surveillance.

These benchmark datasets can be accessed at https://github.com/CDCgov/datasets-sars-cov-2. To ensure consistency, the benchmark data have been downloaded independently onto different computers at multiple institutions and have been confirmed to be identical. Continuous integration with GitHub Actions is maintained, such that whenever the repository changes, a remote computer at GitHub downloads all the datasets and checks them against the hashsums.

Future datasets

As the SARS-CoV-2 pandemic continues, new important lineages are certain to emerge, and additional lineages can be added to these benchmark datasets as needed with the use of version control on GitHub. This platform is also able to accept new benchmark datasets that address scenarios not covered by our current collection, such as replicate sequencing following amplification with varied primer sets. Users can prepare datasets for consideration by copying our spreadsheet formula, adding their own data to it, and submitting it to the GitHub repository via a pull request. More detailed instructions are available in our GitHub repository. Spreadsheets of new benchmark datasets will be evaluated to confirm that all required fields have been provided in the correct format for the GenFSGopher.pl script, and for data integrity via a hashsums for the expected sequence files downloaded from the SRA. If the dataset passes these criteria, then we can use the pull request feature in GitHub to include these new entries in the repository. In the future, some of these checks may be automated using continuous integration GitHub Actions. In this way, we hope that this GitHub repository becomes a central location for SARS-CoV-2 benchmark datasets.

Conclusion

This work describes six benchmark datasets of importance to the global effort in tracking ongoing SARS-CoV-2 evolution and in public health surveillance. These datasets are useful for testing bioinformatics pipelines, both those already established and in production as well as those under development. In addition, these data can be used as a resource for training laboratory personnel and building regional sequencing capacity. Under each of these circumstances, individual datasets can assist users to: test phylogenetic signals, corroborate cross-platform chemistries, confirm SARS-CoV-2 lineage designations, and verify QC thresholds.

By adopting an open access format this effort helps set the stage for future genomic datasets for SARS-CoV-2 to be included in this publicly available benchmarking repository available via GitHub. Additionally, we have also developed a mechanism through GitHub to accept new benchmark datasets through standard repository pull requests, making the repository a central location for benchmark datasets to support SARS-CoV-2 bioinformatic analyses.

Supplemental Information

Table S1 GISAID accessions

Click here for additional data file.

Table S2 The summary of quality control metrics for SARS-CoV-2 benchmark datasets

FastQC, Samtools and TheiaCoV have been used to evaluate key QC metrics listed in Table 2 including quality of raw reads, depth, breath of coverage, quality of resulted assemblies.

Click here for additional data file.

Supplemental Information 3 GISAID accessions and acknowledgements

Click here for additional data file.

We acknowledge Jessica C. Chen at CDC for constructive suggestions on manuscript writing. We also thank all data contributors. The authors of the specimens retrieved from GISAID are named in the Supplemental File. Thank you to all who have made contributions on GitHub through pull requests. Thank you to StaPH-B for creating and maintaining a Docker container and to Peter van Heusden for creating and maintaining a Conda environment.

Additional Information and Declarations

Competing Interests

Author Contributions

Data Availability

The authors declare there are no competing interests.

Lingzi Xiaoli performed the experiments, analyzed the data, prepared figures and/or tables, authored or reviewed drafts of the article, and approved the final draft.

Jill V. Hagey performed the experiments, analyzed the data, prepared figures and/or tables, authored or reviewed drafts of the article, and approved the final draft.

Daniel J. Park performed the experiments, analyzed the data, authored or reviewed drafts of the article, and approved the final draft.

Christopher A. Gulvik performed the experiments, analyzed the data, authored or reviewed drafts of the article, and approved the final draft.

Erin L. Young performed the experiments, analyzed the data, authored or reviewed drafts of the article, and approved the final draft.

Nabil-Fareed Alikhan performed the experiments, analyzed the data, authored or reviewed drafts of the article, and approved the final draft.

Adrian Lawsin performed the experiments, analyzed the data, authored or reviewed drafts of the article, and approved the final draft.

Norman Hassell conceived and designed the experiments, authored or reviewed drafts of the article, and approved the final draft.

Kristen Knipe conceived and designed the experiments, authored or reviewed drafts of the article, and approved the final draft.

Kelly F. Oakeson conceived and designed the experiments, authored or reviewed drafts of the article, and approved the final draft.

Adam C. Retchless conceived and designed the experiments, performed the experiments, analyzed the data, authored or reviewed drafts of the article, and approved the final draft.

Migun Shakya conceived and designed the experiments, performed the experiments, analyzed the data, authored or reviewed drafts of the article, and approved the final draft.

Chien-Chi Lo conceived and designed the experiments, performed the experiments, analyzed the data, authored or reviewed drafts of the article, and approved the final draft.

Patrick Chain conceived and designed the experiments, authored or reviewed drafts of the article, and approved the final draft.

Andrew J. Page conceived and designed the experiments, performed the experiments, analyzed the data, authored or reviewed drafts of the article, and approved the final draft.

Benjamin J. Metcalf conceived and designed the experiments, performed the experiments, analyzed the data, authored or reviewed drafts of the article, and approved the final draft.

Michelle Su conceived and designed the experiments, performed the experiments, analyzed the data, authored or reviewed drafts of the article, and approved the final draft.

Jessica Rowell conceived and designed the experiments, performed the experiments, analyzed the data, authored or reviewed drafts of the article, and approved the final draft.

Eshaw Vidyaprakash conceived and designed the experiments, authored or reviewed drafts of the article, and approved the final draft.

Clinton R. Paden conceived and designed the experiments, authored or reviewed drafts of the article, and approved the final draft.

Andrew D. Huang conceived and designed the experiments, authored or reviewed drafts of the article, and approved the final draft.

Dawn Roellig conceived and designed the experiments, authored or reviewed drafts of the article, and approved the final draft.

Ketan Patel conceived and designed the experiments, authored or reviewed drafts of the article, and approved the final draft.

Kathryn Winglee conceived and designed the experiments, authored or reviewed drafts of the article, and approved the final draft.

Michael R. Weigand conceived and designed the experiments, performed the experiments, analyzed the data, prepared figures and/or tables, authored or reviewed drafts of the article, and approved the final draft.

Lee S. Katz conceived and designed the experiments, performed the experiments, analyzed the data, prepared figures and/or tables, authored or reviewed drafts of the article, and approved the final draft.

The following information was supplied regarding data availability:

The genome sequence accessions are available at GitHub: https://github.com/CDCgov/datasets-sars-cov-2.

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
