# Peer review of "Benchmark datasets for SARS-CoV-2 surveillance bioinformatics"

_PeerJ, doi:10.7717/peerj.13821_

## Round 0.1 · original submission · Major Revisions

Please address all the concerns raised by the reviewers. Once you have addressed them all we will send the manuscript for another round of peer-review.

·

Basic reporting

In viral molecular epidemiological methods development, the existence of "ground truth" datasets with which to accurately benchmark tools is absolutely critical. In the case of past epidemics in which Sanger sequencing was commonly used (e.g. HIV, HCV, Ebola), viral genomic sequences that were obtained were typically quite high-quality, but at the expense of throughput. Throughout the COVID-19 pandemic, however, SARS-CoV-2 genome sequencing has been massively high-throughput, with 10 million genome sequences available just ~2 years since the start of the pandemic (which is amazing), but because of limited Bioinformatics experience from many labs, and because of technical limitations of the sequencers used, many of the genome sequences are not quite trustworthy enough to be assumed to be "ground truth" (which is not so good). In this paper, the authors provide a curated collection of high-quality SARS-CoV-2 sequencing datasets spanning multiple different sequencing technologies, which the authors hope can be used to benchmark bioinformatics methods. The authors provide the curated datasets as well as a toolkit to enable bioinformaticians to query and download the datasets easily. The paper is clear, unambiguous, and well-written. The authors provide sufficient literature references that provide sufficient field background/context. The manuscript has a professional article structure, and the figure and tables are clear. Files and scripts for performing the data scraping are publicly available in a GitHub repository.

Experimental design

The research presented in this manuscript seem to fall within the Aims and Scope of the journal. The research question is well-defined, relevant, and meaningful, and the authors clearly state how this work fills an identified knowledge gap (lack of high-quality benchmark SARS-CoV-2 genomic datasets). The investigation was performed at a high technical standard, and all datasets utilized were public, so there were no concerns regarding ethical standards. The methods describe with sufficient detail and information to replicate inclusion/exclusion filtering, and the GitHub repository contains sufficient information to install and utilize the tools the authors developed to pull the datasets.

My main challenge when trying to run the tool was that, for such a simple task ("pull curated datasets"), I had to install the toolkit the authors developed, which was much less trivial than I was expecting. When I read the abstract, I was expecting links to public HTTP/FTP servers from which I could just smoothly pull data using wget or similar, but one needs to install the authors' toolkit which itself depends on the SRA Toolkit and some NCBI tools (e.g. edirect). The authors provide a Docker/Singularity image as well as a BioConda environment, so it's not too bad for folks who already have those tools installed, but it seems a bit overkill for a task as simple as "I want to download this curated dataset the authors have described", which I assumed I would be able to do with a single wget command or similar. I understand the desire for this more complex toolkit, as it does more than simply pull a given dataset and future updates could expand which datasets are included, but it would be nice if "releases" of the curated datasets could be posted somewhere from which folks could directly pull via wget/similar.

Regardless, I was unable to install the tool from scratch because I gave up trying to install the dependencies, but I was indeed able to run it successfully from the Docker image precisely as described, and the tool worked simply and quickly.

Validity of the findings

The inclusion/exclusion criteria are sound, the technical details of the Bioinformatics analyses described for determining inclusion/exclusion criteria are sound, and all data/scripts are publicly accessible. I was able to run the tool as described successfully (via the Docker image the authors provide).

·

Basic reporting

Xiaoli and collaborators created both a tool and a set of different datasets that can be used to standardize the benchmark step in studies of workflow development of SARS-CoV-2 genome assembly. They organized six different datasets covering the most-used sequencing platforms used in the genomic surveillance of SARS-CoV-2 - Nanopore and Illumina - and which addresses different bioinformatics problems. Moreover, the authors worked considering the good practices of software development and made the code publicly available, as well as created different ways of implementation - source code, conda, docker, singularity - which aims to attend research groups with different computational environments and last but not least, created user-friendly documentation on GitHub, which allows non-expert users to use their tool and recover the datasets. Besides the excellent quality of the tool and manuscript, I recommend the authors improve some points in the manuscript as well as the GitHub documentation in a way to clarify some of the points addressed in the ‘Experimental design’ and ’General comments’ sections.

Experimental design

Recent publications on the development of bioinformatics workflows focused on SARS-CoV-2 genome assemblies, such as poreCov, HAVoC, ViralFlow, and PipeCov, used different datasets of SARS-CoV-2 sequencing reads. Xiaoli and collaborators raised in their manuscript the importance of creating benchmark datasets to improve the development of the above-cited workflows. The manuscript is well written, with a logical order of introduction of the field, the problem that should be solved, what was done to solve the problem, and the description of the results.

Evaluating the GitHub repository of the study, I noticed that the authors were careful in both the tool and dataset documentation, but some points could be improved (see General comments). I tested the three modes of implementation (I can’t test the source code method owing to an error during the edirect implementation, following the instructions present on GitHub, the edirect dir doesn’t have a setup.sh script (edirect_dir.txt on attachments)): conda (Ubuntu 20.04 LTS, Python 3.8.10, Conda 4.11.0); docker (Ubuntu 20.04 LTS, Docker version 20.10.7, build 20.10.7-0ubuntu5~20.04.2) and singularity (Ubuntu 20.04 LTS, Singularity v.3.8.0), and the three modes don’t show any error to the implementation, great work. Besides the good documentation, I noticed some incongruences (see the review_test.zip files in attachments) between the conda and the docker/singularity runs, for example, in the container modes, the directories of SRA were not removed at the end of the analysis and the *.fna files were empty (docker_run_dir.txt and sing_run_dir.txt), as well as both docker and singularity runs informs and error after download the fastq files (GenFSGopher.pl: ERROR: `make` failed. Please address all errors and then run the make command again: nice make all --directory=/data/<dataset> --jobs=12 Died at /home/user/datasets-sars-cov-2/scripts/GenFSGopher.pl line 523), could those errors be a local problem or a problem related to the docker and singularity modes?

After reading the manuscript and the GitHub documentation, it is clear that GenFSGopher.pl Perl script performs the described analysis, but the set of python scripts (as cited in lines 223-228, and represented in Figure 1) are called for which purpose? they have been involved in GenFSGopher.pl (I can’t see it inside the code). In my opinion, the authors should clarify this question in the text, as well as should split the Figure 1 into two sections: A - with the workflow scheme of GenFSGopher.pl and B - The current workflow schema present in Figure 1 about the ‘Automated workflow for identifying representative sequences for datasets’. Moreover, where are those in-house python scripts available? I don’t find them in the git repository.

In the methods section, the authors described in the ‘Sequence quality evaluation’, the evaluation of several metrics using the Titan, and through Tables 1 and 2 the authors inform which steps are performed, and which are the threshold values. Where are the results of these analyses? Maybe this is the main point that should be addressed to follow with the review process. If the main goal of this study is to provide a benchmark dataset to be used in the development of bioinformatics tools/workflows/infrastructures, in my point of view, at least the results of the quality and general metrics of sequences present in each dataset (expected coverage depth, breadth of coverage, SNPs and indels, lineages) should be informed in the results section as well as as a material in the GitHub repository. I know that several of that information can be recovered using the EPI code on GISAID, but the idea is to show the benchmark of a tool, or different tools, using these benchmark datasets present in the manuscript. In brief, both the tool and datasets are well described. Tables 1 and 2 are in line with the text, but some points such as the improvement of Figure 1, and a Results section exploring the benchmark of tools using those benchmark datasets should be performed.

Validity of the findings

Considering the tool documentation, and the results of the manuscript, the benchmark datasets can be easily accessed using the developed tool with the provided dataset tsv files. The authors concluded the study in line with the raised problem and their results. One interesting point, as the authors discussed in the last paragraph of the ‘results and discussion section’, is that this tool can be used to create new benchmark datasets, it is very important considering the latest studies about co-infected samples and recombinant lineages.

Additional comments

To improve both the quality of the manuscript and GitHub documentation, I recommend to authors address some specific points:

In the manuscript

lines 78-91 - Could the authors provide the references that support the statements in this section?
125-126 - I think that the authors could move the “A summary of the workflow used to identify and validate sequences to be included in the datasets is provided in Figure 1.” to the “Download script” section and merge this section with “Automated data mining workflow”. In my opinion, is clear that the study has 2 major parts: The creation of the tool, and the establishment of the six datasets of SARS-CoV-2, so, why the “Download Script” and “Automated data mining workflow” sections are split, they both don’t refer to the development of the tool?
166-182, 197-207, 209-221 - For the dataset 1 and 4, the authors described the sequencing method/protocols used to generate the datasets, but for the datasets 2, 3, 5, and 6, not, could the authors add this information in the text?.
189 - “e VOIs (B.1.525, B.1.526, P.2, B.1.617.1, B.1.617.2)”, B.1.617.2 is a VOC.
223-254 - As pointed out in the ‘Experimental Design’, where are those python scripts, and how are they related to the GenFSGopher.pl?
Figure 1 - This figure is related to different information during the text: Summary of Workflow to recovery dataset (line 126), Sequence quality (143) Automated workflow (225). As mentioned in the Experimental design, I recommend the authors to split this information, in both Figure 1 and text, to organize the idea in a logical order: The GenFSGopher.pl steps schema; The extra steps schema (Titan and “in-house” python scripts).
Results - I recommend the authors include the results obtained with Titan, and organize a table, or a supplementary table, with the general information cited in the ‘Experimental Design’ section in the manuscript.

On the GitHub documentation

Summary Table: Add two new columns: One with the tsv name of the dataset files, and another with the link to the primer set (in case of amplicon sequencing datasets). The second column will provide key information for groups that work with the development of amplicon sequencing amplicons, considering that the primer information should be parsed before (hard-clip) or after (soft-clip) mapping step.
Install & Usage section:
Add a tip to the user to identify the conda path, such as “whereis” for example;
Add Docker and Singularity versions that the tool was tested on.
Considering the quality control step described in the manuscript, if a specific SRA accession doesn’t pass the quality criteria, the fastq files won’t be recovered right? Maybe a list of which files should be expected to recover at the end of the run of each dataset could improve the reproducibility of the tool as well as allow the user to identify if something got wrong.
Create a new material with general information (expected coverage depth, breadth of coverage, SNPs and indels, lineages) related to the expected fastq files.
Fix the Source code implementation section, at least in the part of edirect compilation.
As the authors cited, new lineages will originate in the course of pandemic, and future benchmark datasets will be necessary. I know that it depends on the submission to SRA by different groups, but datasets of co-infected and/or recombinant samples will be extremely hopeful to benchmark new workflows.

Reviewer 3 ·

Basic reporting

Great introduction with a clear description of the current landscape for SARS-CoV-2 genomic surveillance.
Extremely well-written manuscript over all.

Experimental design

Excellent documentation of the datasets on github link with very clear instructions for use.

I recommend adding the in-house scripts used in the automated data mining workflow to the github site for others to see exactly how these steps were performed so steps can be repeated for similar use.

Validity of the findings

The authors of this manuscript provide a valuable curation of datasets for SARS-CoV-2 bioinformatics tool quality control and benchmarking. This is a key resource for public health and other communities performing SARS-CoV-2 surveillance. Vital quality control references, such as these datasets, fill a much-needed gap that has existed since the beginning of the pandemic.

Additional comments

A few references need to be corrected to complete the citation in the Reference sections (i.e., provide links to access source).
1. Muthukadan B. 2018 Selenium
2. Seemann T. 2019. Snippy
3. Titan. 2021

---

## Round 0.2 · accepted · Accept

Thanks for addressing all the concerns of the reviewers.

·

Basic reporting

Xiaoli and collaborators created both a tool and a set of different datasets that can be used to standardize the benchmark step in studies of workflow development of SARS-CoV-2 genome assembly. They organized six different datasets covering the most-used sequencing platforms used in the genomic surveillance of SARS-CoV-2 - Nanopore and Illumina - and which addresses different bioinformatics problems. Moreover, the authors worked considering the good practices of software development and made the code publicly available, as well as created different ways of implementation - source code, conda, docker, singularity - which aims to attend research groups with different computational environments and last but not least, created user-friendly documentation on GitHub, which allows non-expert users to use their tool and recover the datasets.

Experimental design

Recent publications on the development of bioinformatics workflows focused on SARS-CoV-2 genome assemblies, such as poreCov, HAVoC, ViralFlow, and PipeCov, used different datasets of SARS-CoV-2 sequencing reads. Xiaoli and collaborators raised in their manuscript the importance of creating benchmark datasets to improve the development of the above-cited workflows. The manuscript is well written, with a logical order of introduction of the field, with the last changes in the manuscript the methods are well described in the light of reproducibility.

Validity of the findings

Considering the tool documentation, and the results of the manuscript, the benchmark datasets can be easily accessed using the developed tool with the provided dataset tsv files. The authors concluded the study in line with the raised problem and their results, both tool and repository can be used to create and make available new datasets.

Additional comments

I would like to thank the authors for the improvement in both manuscript text and Github repository documentation after the first round of revision. This work certainly will help - as is it already helping - groups that work with bioinformatics and epidemiology around the globe.

Reviewer 3 ·

Basic reporting

Thank you for adding the python scripts to the github repository in response to the previous comment.

Experimental design

No comment.

Validity of the findings

No comment.

Additional comments

The reference for Titan has been corrected; however, links still need to be provided for the following two references in the reference section.

Muthukadan B. 2018 Selenium
Seemann T. 2019. Snippy

This is minor and can be addressed during the proof review.